# SALIENT OBJECT RANKING VIA CYCLICAL PERCEPTION-VIEWING INTERACTION MODELING

**Rongjin Guo,  Ke Xu,**[*]  **Rynson W.H. Lau**[*]
Department of Computer Science, City University of Hong Kong
Rongjin.Guo@my.cityu.edu.hk, kkangwing@gmail.com,
Rynson.Lau@cityu.edu.hk

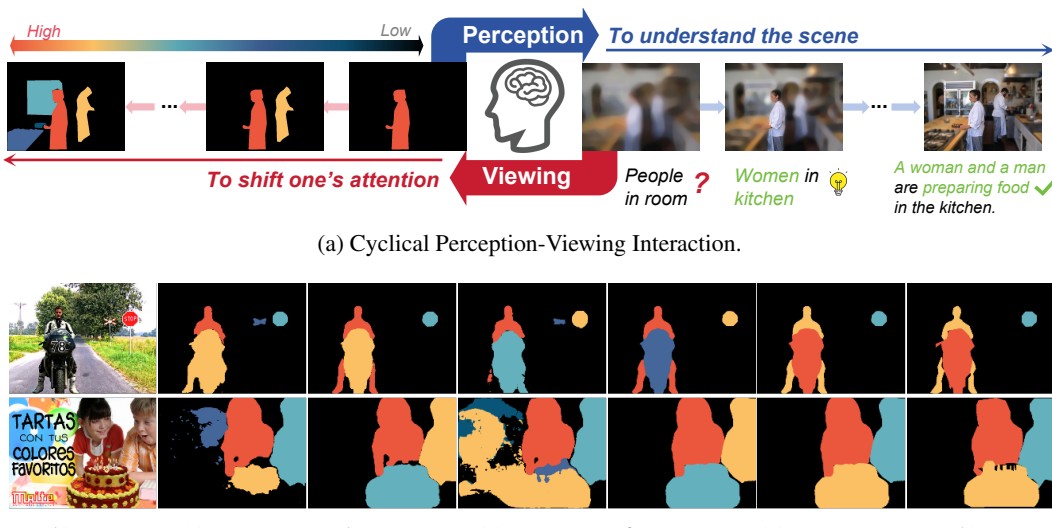

(a) Cyclical Perception-Viewing Interaction.

| (b) Input | (c) SeqRank | (d) DSGNN | (e) PoseSOR | (f) Ours (initial) | (g) Ours (refined) | (h) GT |

Figure 1: Illustration of our cyclical interaction framework (a), where perception and viewing alternately influence each other, guiding attention shifts and deepening scene comprehension. Existing state-of-the-art methods, such as (c) to (e), often fail in semantically rich scenes due to their heavy reliance on low-level visual cues. Our initial prediction (f) exhibits similar limitations. However, through the proposed iterative refinement approach, our model progressively corrects these errors by modeling cyclical perception-viewing interactions, leading to accurate saliency rankings (g).

## ABSTRACT

Salient Object Ranking (SOR) aims to predict human attention shift across different salient objects in a scene. Although a number of methods have been proposed for the task, they typically rely on modeling the bottom-up influences of image features on attention shifts. In this work, we observe that when free-viewing an image, humans instinctively browse the objects in such a way as to maximize contextual understanding of the image. This implies a cyclical interaction between content (or story) understanding of the image and attention shift over it. Based on this observation, we propose a novel SOR approach that models this explicit top-down cognitive pathway with two novel modules: a story prediction (SP) module and a guided ranking (GR) module. By formulating content understanding as the image caption generation task, the SP module learns to generate and complete the image captions conditioned on the salient object queries of the GR module, while the GR module learns to detect salient objects and their viewing orders guided by the SP module. Extensive experiments on SOR benchmarks demonstrate that our approach outperforms state-of-the-art SOR methods. Codes are available here.

---

[*]Corresponding authors.

# 1 INTRODUCTION

Salient Object Ranking (SOR) aims to model the human attention shift across salient objects in a scene, by detecting and ranking a sequence of salient objects with their corresponding viewing orders. SOR can facilitate human visual behavior understanding (Lin et al., 2024) and various downstream computer vision tasks, such as scene understanding (Li et al., 2023; Du et al., 2019; Zhang et al., 2014) and autonomous driving (Huang & Wang, 2024).

Islam et al. (2018) first propose to rank the saliency degrees of objects by predicting a relative saliency map based on the consensus degrees of multiple viewers. Later, Siris et al. (2020) propose the SOR task to study how humans shift their attentions across salient objects, with a neural method to predict saliency ranks based on modeling the relation between objects and global contexts. Liu et al. (2021a) propose a graph-based network to learn relations among objects and local contexts. Tian et al. (2022a) propose to leverage both spatial and object-based attention mechanisms, which are used in the human visual system, to model the bidirectional object-context relations. Recently, Guan & Lau (2024b) propose to model the processing of visual information and attention shifts in the human visual system by incorporating both foveal and peripheral vision for SOR. Qiao et al. (2024) propose to model the impact of scene context on attention shifts by constructing a scene graph to reason the saliency ranks. Guan & Lau (2024a) demonstrate that human pose affects the observer's attention, and propose to incorporate human poses to infer SOR. All these existing methods primarily consider bottom-up factors (i.e., visual features and semantic information) that influence human attention (Ramos Gameiro et al., 2017).

In this work, we observe that when given an image for free viewing, our brain instinctively engages in scene perception to maximize contextual understanding (Murlidaran & Eckstein, 2024; Rayner & Pollatsek, 1992), with fixations concentrated on objects that are critical for comprehending the overall scene (Murlidaran & Eckstein, 2024). This process can be viewed as a *cyclical interaction between scene perception and eye movement*, studied as active perception and predictive processing in cognitive science (Rao & Ballard, 1999; Peelen et al., 2024; Zacks et al., 2007; Berman & Colby, 2009). In other words, human attentions are continuously shifted across salient objects, driven by the evolving scene-level understanding. The observer first focuses on objects that are essential for understanding the scene and potentially forms predictions about the story behind the scene. This perceptual process then guides the observer's attention shift, which in turn shapes the predicted story as it moves through the scene. The attention continues shifting until the final salient object is reached, at which point the predicted story stabilizes and the observer gains a more complete understanding of the image content, as illustrated in Fig. 1(a).

Motivated by the above observation, we propose in this paper a novel object query-based method to model the cyclical interaction of human perception and attention shift for SOR. Our method has two novel modules, the *story prediction (SP) module* and the *guided ranking (GR) module*. By formulating the "contextual understanding" as the image caption generation process, the SP module predicts and refines the image caption conditioned on the current saliency ranking result from the GR module. Meanwhile, the GR module learns to refine the saliency ranks while incorporating the text modality from the SP module as guidance. The SP and GR modules perform synergistically to summarize the input image through caption generation while predicting the attention shift across salient objects. As shown in Fig. 1(b), our method can self-refine its predictions through the cyclical perception-viewing interaction, producing more faithful ranking results compared to the state-of-the-art methods. In summary, we make the following main contributions:

1. We propose a novel approach that incorporates a top-down cognitive process (scene understanding), which is inspired by psychological studies, for SOR. The key idea is to model an explicit cyclical interaction between content perception and human attention shift in free-viewing.

2. Our SOR approach has two novel modules: a story prediction (SP) module and a guided ranking (GR) module. The SP module simulates the brain's process of predicting the story behind the scene using a generative captioning model, conditioned on the latest predicted saliency ranks, while the GR module predicts the saliency ranks by iteratively refining object queries, guided by the latest predicted story.

3. Extensive experiments demonstrate the effectiveness of our model, and that our method outperforms the state-of-the-art SOR methods.

## 2 RELATED WORK

**Salient Object Ranking (SOR).** When viewing an image, humans typically shift their attentions across salient objects sequentially. Islam et al. (2018) make the first attempt to rank saliency degrees based on the consensus degrees of several observers, which ignores the visual/spatial relations of objects in the scene. Following psychological and behavioral studies (Itti & Koch, 2000; Neisser, 2014), Siris et al. (2020) propose the SOR task to study human attention shift across objects in an image, and a neural network to model the relations between objects and the global image context for SOR. Some SOR methods are subsequently proposed to enhance the SOR performance by incorporating object position coordinates (Fang et al., 2021a), modeling inter-object relations via neural graphs (Liu et al., 2021a), and incorporating both object-based and spatial attention mechanisms (Tian et al., 2022a).

Recently, Guan & Lau (2024b) propose to model sequential viewing and attention shifting by using foveal vision to focus on an object and peripheral vision to locate the next object. Qiao et al. (2024) propose a hyper-graph-based network, while Deng et al. (2024) propose a tri-tiered nested Graph Neural Network, to incorporate object-context relationships, for SOR. Wu et al. (2024) construct a graph for each scene, while explicitly using the shape and texture features of objects as graph edges for SOR prediction. Guan & Lau (2024a) propose to model human poses as cues to help enhance SOR performances. Most recently, Liu et al. (2025) exploit the implicit orders in the descriptions of Large Vision-Language Models (LVLMs) to guide SOR.

All the above works rely on bottom-up image features as cues (i.e., poses, scene contexts, object attributes, and inter-object relations) for SOR predictions, which may not be reliable enough to faithfully reproduce human attention shifts over salient objects. In this work, we propose to incorporate the top-down cognitive process as guidance for SOR, by explicitly modeling the cyclical interaction between image content perception and human attention shifts.

**Salient Object Detection (SOD).** This task aims to identify the most visually conspicuous objects in an image, and has been extensively studied. Early SOD methods (Cheng et al., 2014; Klein & Frintrop, 2011; Perazzi et al., 2012; Achanta et al., 2009) primarily rely on low-level image features, such as contrast, edge, and structure responses, to construct saliency maps. Subsequently, a large number of deep learning based SOD methods (Liu et al., 2021b; Siris et al., 2021; Wang et al., 2023; Wei et al., 2020; Zhang et al., 2019; Veksler, 2023; Tian et al., 2023; Li et al., 2024) are proposed. Their models typically incorporate multi-scale feature fusion (Liu et al., 2021b; Wang et al., 2023), contextual semantic aggregation (Siris et al., 2021; Zhang et al., 2019), multi-tasking (Wang et al., 2018; Zhang et al., 2019; Wei et al., 2020; He et al., 2017b), and attention mechanisms (Liu et al., 2018; Zhang et al., 2018), to enhance spatial coherence and semantic awareness. However, as SOD methods neither differentiate salient instances of the same class nor do they estimate the attention shift across objects, they cannot be directly applied to the SOR task.

**Salient Instance Detection (SID).** It aims to identify each salient object at the instance level. Early SID methods (Fan et al., 2019; Wu et al., 2021) are predominantly based on the Mask R-CNN (He et al., 2017a) architecture. They first detect object instances through region proposal networks and then learn discriminative features to distinguish salient instances from non-salient ones using pixel-level supervision. To reduce the reliance on costly pixel-wise annotated masks for training, some recent works tend to propose weakly-supervised (Tian et al., 2020; 2022b) or unsupervised (Tian et al., 2024) approaches. While SID methods can provide instance-wise segmentation of salient objects, they do not attempt to predict the attention shift across these objects.

## 3 OUR METHOD

In this work, we observe that scene perception can significantly influence visual behavior during free viewing, while attention in turn shapes how our brain understands a scene, indicating that modeling of perception-viewing cycle can help facilitate the understanding of how our visual attention mechanisms operate in free-viewing real-world scenes. Inspired by this, we propose a novel multi-task cyclical learning framework that synergizes Story Prediction (SP) and Guided Ranking (GR) to emulate this cognitive process for Salient Object Ranking (SOR).

Section 3.1 introduces our overall architecture that integrates image captioning (from SP) and saliency ranking (from GR). The Story Prediction (SP) module (Section 3.2) implements contextual

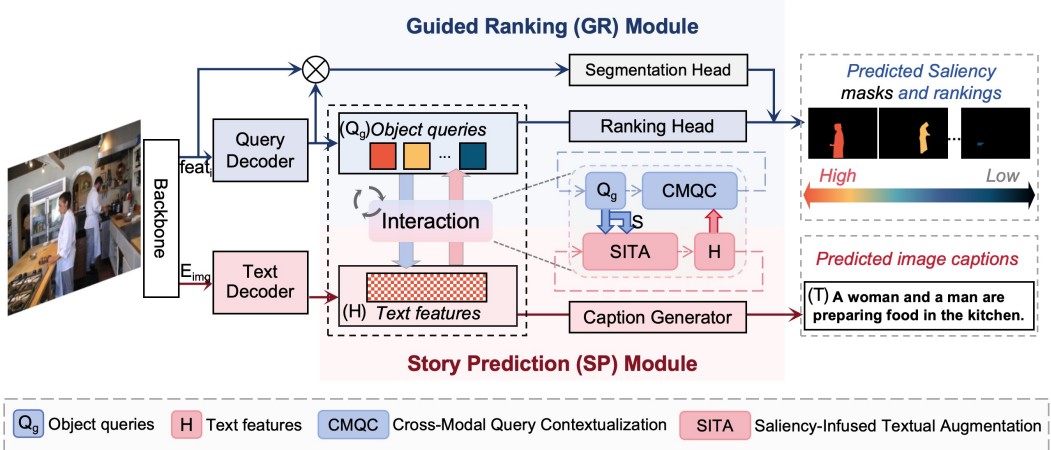

Figure 2: **Overview of our proposed architecture.** This framework comprises two key novel modules: the Guided Ranking (GR) Module and the Story Prediction (SP) Module. The GR Module generates saliency masks and rankings via object queries and cross-modal interactions, while the SP Module produces image captions by integrating text features with Object queries through the Story Prediction process. During the Interaction process, the two modules collaborate and iteratively update the visual and textual representations, achieving collaborative enhancement of visual saliency ranking and image caption generation.

understanding as a dynamic caption generation process, where saliency degree features from GR iteratively influence the refinement of the caption. Concurrently, the Guided Ranking (GR) module (Section 3.3) learns to predict the focused salient instance at each step, explicitly incorporating the text modality generated by SP as guidance to determine subsequent salient instances. Through simultaneous optimization, SP and GR jointly perform the SOR task by generating captions and estimating viewing orders. Finally, Section 3.4 details our training strategy.

## 3.1 OVERVIEW

As shown in Fig. 2, we propose a dual-branch framework that achieves joint reasoning of saliency rank and image description through iterative interaction between object queries and text features. The framework consists of a vision backbone, a Query Decoder, a Text Decoder, a Guided Ranking (GR) module, a Story Prediction (SP) module, and the interaction between the GR and SP modules.

The input image is first fed into the backbone to extract a set of image features. These features are then used to construct a feature pyramid $feats_i \in \mathbb{R}^{C_i \times H_i \times W_i}, i \in \{2, 3, 4, 5\}$ via a feature extractor (Cheng et al., 2022), where $C_i, H_i, W_i$ denote the number of channels, height, and width of the $i$-th feature map, respectively. We initialize a set of object queries $\mathbf{Q}_0 \in \mathbb{R}^{N \times D}$ with learnable parameters, where $N$ and $D$ represent the number and dimension of the queries. A transformer-based decoder with $L$ layers is employed to iteratively enhance the query representations by attending to multi-scale object features:

$$\mathbf{Q}_l = \text{QueryDecoder}(\mathbf{Q}_{l-1}, feats_i), \quad i = (l-1) \bmod 3 + 3, \tag{1}$$

where $l \in \{1, 2, \dots, L\}$ and $\mathbf{Q}_l$ represents output object queries after $l - th$ query decoder layers. The multi-layer object queries are then aggregated to obtain the global query $\mathbf{Q}_g \in \mathbb{R}^{N \times D}$, which are fed into the ranking head to predict saliency scores $\mathbf{S} \in \mathbb{R}^{N \times 1}$:

$$\mathbf{S} = \text{Linear}(\mathbf{Q}_g). \tag{2}$$

Meanwhile, the backbone visual features are projected to obtain image embeddings $E_{\text{img}} \in \mathbb{R}^{N_p \times D_t}$, where $N_p$ denotes the number of image patches. The embeddings $E_{\text{img}}$ align with the dimensions of the textual space $D_t$, and serve as the cross-modal context for autoregressive text generation. Starting from a special "[BOS]" token, the decoder autoregressively generates caption tokens. At each step $t$, it takes as input the embeddings of previously generated tokens, $\mathbf{x}_{<t} = [\mathbf{x}_0, \dots, \mathbf{x}_{t-1}] \in \mathbb{R}^{t \times D_t}$, along with the image embeddings $E_{\text{img}}$, to produce the current hidden state $\mathbf{h}_t \in \mathbb{R}^{D_t}$:

$$\mathbf{h}_t = \text{TextDecoder}(\mathbf{x}_{<t}, E_{\text{img}}). \tag{3}$$

After generating all $L_s$ tokens, we obtain $\mathbf{H} = [\mathbf{h}_1; \ldots; \mathbf{h}_{L_s}] \in \mathbb{R}^{L_s \times D_t}$ as the full text features, which can be projected to the vocabulary space to generate a descriptive text $\mathbf{T}$, as:

$$\mathbf{T} = \arg \max_{w \in Vocab} \text{softmax}(\text{Linear}(\mathbf{H})). \tag{4}$$

**Cyclic Interaction.** We establish a cyclic interaction between $\mathbf{Q}_g$ and $\mathbf{H}$ through two modules: the Guided Ranking (GR) module and the Story Prediction (SP) module. Our SP module learns to enhance text features via the Saliency-Infused Textual Augmentation (SITA), and then generate descriptive language by a caption generator. The enhanced text features $\mathbf{H}^{(k)}$ can be computed as:

$$\mathbf{H}^{(k)} = \text{SITA}(\mathbf{Q}_g^{(k-1)}, \mathbf{S}^{(k-1)}, \mathbf{H}^{(k-1)}), \tag{5}$$

where $k$ is the index of iterations. Concurrently in the GR module, the enhanced text features $\mathbf{H}^{(k)}$ contextualize the global query $\mathbf{Q}_g$ through the Cross-Modal Query Contextualization (CMQC) mechanism, allowing it to learn image-perception features, as:

$$\mathbf{Q}_g^{(k)} = \text{CMQC}(\mathbf{Q}_g^{(k-1)}, \mathbf{H}^{(k)}). \tag{6}$$

This iterative process is repeated for $K$ steps. The final object query $\mathbf{Q}_g^{(K)}$ is fed into a ranking head to predict saliency scores, as described in Eq. 2. Meanwhile, the final text features $\mathbf{H}^{(K)}$ are decoded into natural language captions through a generator, following Eq. 4. The overview of SITA and CMQC are shown in Fig. 3.

## 3.2 STORY PREDICTION (SP) MODULE

We propose the Story Prediction (SP) module to simulate the brain's perception process by formulating scene understanding as generating image captions. The SP module establishes the viewing-to-perception pathway, where saliency-related information influences the generation of linguistic descriptions. Specifically, it iteratively injects object query features into text features, enabling the model to ground textual narratives in visually salient regions.

The SP module consists of a Saliency-Infused Textual Augmentation (SITA) module for modality-aligned feature fusion and a caption generator to progressively align linguistic descriptions with human attention patterns. The SITA module takes the global query $\mathbf{Q}_g^{(k-1)} \in \mathbb{R}^{N \times D}$ and its associated saliency scores $\mathbf{S}^{(k-1)} \in \mathbb{R}^{N \times 1}$ as inputs (Eq. 5). SITA first computes a saliency-weighted visual context vector through an element-wise multiplication of queries with saliency scores and then a spatial averaging along the object dimension, yielding a compact visual representation $\mathbf{V}_{\text{sal}} \in \mathbb{R}^D$, as:

$$\mathbf{V}_{\text{sal}} = \frac{1}{N} \sum_{i=1}^{N} (\mathbf{Q}_g[i] \odot \mathbf{S}[i]), \tag{7}$$

where $\odot$ denotes element-wise multiplication. This vector $\mathbf{V}_{\text{sal}}$ is then projected to align with the text feature dimension $D_t$ and broadcast to match the text sequence length $L_s$, resulting in $\mathbf{V}_{\text{sal}}^{\text{align}} \in \mathbb{R}^{L_s \times D_t}$.

These expanded features $\mathbf{V}_{\text{sal}}^{\text{align}}$ are then processed by a gating mechanism, which is structured as a two-layer neural network with GELU activation

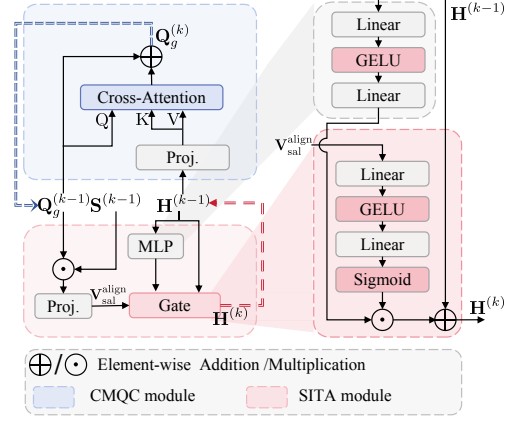

Figure 3: **Cross-Modal Query Contextualization (CMQC) module and Saliency-Infused Textual Augmentation (SITA) module.** CMQC contextualizes object queries with text semantics via cross-modal attention, while SITA injects saliency-guided visual cues into text features through adaptive gating. Jointly, they enable iterative refinement of visual-language representations for saliency ranking and caption generation.

and sigmoid normalization. The generated gate $\mathbf{G} \in \mathbb{R}^{L_s \times 1}$ dynamically scales the outputs of a MLP network applied to the original text features. A residual connection preserves the baseline linguistic patterns while allowing controlled infusion of saliency information. The whole process can be described as:

$$\mathbf{G} = \sigma(\text{GELU}(\mathbf{V}_{\text{sal}}^{\text{align}} \mathbf{W_1} + \mathbf{b_1}) \mathbf{W_2} + \mathbf{b_2}), \tag{8}$$

$$\mathbf{H}^{(k)} = \text{MLP}(\mathbf{H}^{(k-1)}) \odot \mathbf{G} + \mathbf{H}^{(k-1)}. \tag{9}$$

This gating mechanism is cognitively inspired, which mimics neural gain modulation (Peelen et al., 2024) by adaptively scaling textual features based on saliency information, aligning with attentional modulation theories while ensuring saliency-driven augmentation of text features (see Appendix A.1).

### 3.3 GUIDED RANKING (GR) MODULE

The Guided Ranking Module predicts object viewing order by enhancing the object queries through a perception-to-viewing pathway with two key components: (1) the Cross-Modal Query Contextualization (CMQC) that iteratively refines query representations using linguistic features, and (2) a ranking head that outputs saliency scores after refinement iterations.

The CMQC module first projects high-dimensional text features into a latent space commensurate with object query embeddings. Given input text features $\mathbf{H} \in \mathbb{R}^{L_s \times D_t}$ from the text decoder, a learnable linear transformation with layer normalization adapts them to the query dimension ($\mathbb{R}^D$), enabling cross-modal interaction while preserving the linguistic structure via normalization constraints.

Subsequently, we employ a multi-head cross-attention mechanism to iteratively refine semantic representations across $K$ steps. In each iteration $k$, object queries $\mathbf{Q}_g^{(k)} \in \mathbb{R}^{N \times D}$ interact with text features via scaled dot-product attention, enabling targeted alignment with relevant linguistic cues (e.g., associating clothing-related queries with tokens like "striped shirt"). The attention weights adaptively fuse contextualized textual semantics into the queries through residual updates:

$$\mathbf{Q}_g^{(k+1)} = \mathbf{Q}_g^{(k)} + \text{MultiHeadAttn}(\mathbf{Q}_g^{(k)}, \mathbf{H}^{(k)}). \tag{10}$$

This residual architecture preserves spatial priors while progressively integrating cross-modal semantics. Over $K$ iterations, the queries evolve to emphasize contextual features while suppressing irrelevant linguistic noise. This iterative refinement process is analogous to predictive coding in the cognitive system (Rao & Ballard, 1999), where the residual update minimizes the prediction error between object queries and their expected values under textual guidance (see Appendix A.2). The final saliency ranking scores are computed following Eq. 2.

### 3.4 TRAINING LOSS

Our model is trained in an end-to-end manner with the loss function consisting of four terms, as:

$$\mathcal{L} = \mathcal{L}_{task} + \mathcal{L}_{rank} + \mathcal{L}_{lm}. \tag{11}$$

$\mathcal{L}_{task}$ follows the loss configuration of Mask2Former (Cheng et al., 2022). It includes $\mathcal{L}_{mask}$ for predicting instance masks and $\mathcal{L}_{cls}$ for determining whether each instance is a salient object, as:

$$\mathcal{L}_{task} = \mathcal{L}_{mask} + \mathcal{L}_{cls}, \tag{12}$$

where $\mathcal{L}_{mask}$ adopts the binary cross-entropy loss and the dice loss. $\mathcal{L}_{cls}$ adopts the cross-entropy loss. $\mathcal{L}_{rank}$ is the saliency ranking loss (Liu et al., 2021a). $\mathcal{L}_{lm}$ is the cross-entropy loss to compute the difference between the generated and the ground truth image captions.

## 4 EXPERIMENTS

**Implementation Details.** We employ a Swin Transformer pre-trained on the MS-COCO (Lin et al., 2014) training set as the backbone for feature extraction, and a pre-trained BLIP's text decoder to generate $\mathbf{H}^{(0)}$. For each image, we randomly select one out of its five corresponding captions from the MS-COCO dataset as the ground-truth caption. Our model is initialized with configuration parameters $N$=200, $K$=5, and $D$=256, trained end-to-end without layer freezing across four RTX 3090 GPUs, with all input images resized to a 1024×1024 resolution. We employ the AdamW optimizer with a 1e$^{-4}$ weight decay, and train our model for 24,000 iterations with a batch size of 4. The learning rate is initially set to 2.5e$^{-5}$ and reduced by 10 after 14,000 iterations. During inference, objects with confidence scores over 0.7 are regarded as salient ones for follow-up ranks prediction.

**Evaluation Datasets.** We conduct experiments on the publicly available SOR benchmark datasets, ASSR (Siris et al., 2020) and IRSR (Liu et al., 2021a). The ASSR dataset contains 7,646 training

Table 1: Quantitative Comparison. SOD: Salient Object Detection task. SID: Salient Instance Detection task. INS: Instance Segmentation task. SOR: Salient Object Ranking task. Best results are marked in **bold** and second-best results are underlined. '-' indicates that the result is not available.

| Methods | Venues | Tasks | Backbone | ASSR Dataset | | | IRSR Dataset | | |
|---|---|---|---|---|---|---|---|---|---|
| | | | | SA-SOR ↑ | SOR ↑ | MAE ↓ | SA-SOR ↑ | SOR ↑ | MAE ↓ |
| S4Net (Fan et al., 2019) | CVPR-2019 | SID | ResNet-50 | 0.451 | 0.649 | 14.4 | 0.224 | 0.611 | 12.1 |
| VST (Liu et al., 2021b) | ICCV-2021 | SOD | T2T-ViT-T | 0.422 | 0.643 | 9.99 | 0.183 | 0.571 | 8.75 |
| MENet (Wang et al., 2023) | CVPR-2023 | SOD | ResNet-50 | 0.369 | 0.627 | 9.60 | 0.162 | 0.558 | 8.25 |
| QueryInst (Fang et al., 2021b) | ICCV-2021 | INS | ResNet-101 | 0.596 | 0.865 | 8.52 | 0.538 | 0.816 | 7.13 |
| Mask2Former (Cheng et al., 2022) | CVPR-2022 | INS | ResNet-101 | 0.635 | 0.867 | 7.31 | 0.521 | 0.799 | 7.14 |
| RSDNet (Islam et al., 2018) | CVPR-2018 | SOR | ResNet-101 | 0.386 | 0.692 | 18.2 | 0.326 | 0.663 | 18.5 |
| ASRNet (Siris et al., 2020) | CVPR-2020 | SOR | ResNet-101 | 0.590 | 0.770 | 9.39 | 0.346 | 0.681 | 9.44 |
| PPA (Fang et al., 2021a) | ICCV-2021 | SOR | VoVNet-39 | 0.635 | 0.863 | 8.52 | 0.521 | 0.797 | 8.08 |
| IRSR (Liu et al., 2021a) | TPAMI-2021 | SOR | ResNet-50 | 0.650 | 0.854 | 9.73 | 0.543 | 0.815 | 7.79 |
| OCOR (Tian et al., 2022a) | CVPR-2022 | SOR | Swin-L | 0.541 | **0.873** | 10.2 | 0.504 | 0.820 | 8.45 |
| PSR (Sun et al., 2023) | ACMMM-2023 | SOR | ResNet-50 | 0.644 | 0.815 | 9.59 | 0.454 | 0.752 | 8.07 |
| HyperSOR (Qiao et al., 2024) | TPAMI-2024 | SOR | ResNet-101 | 0.653 | 0.830 | 10.01 | - | - | - |
| SeqRank (Guan & Lau, 2024b) | AAAI-2024 | SOR | Swin-L | 0.663 | 0.863 | 8.03 | 0.554 | 0.801 | 7.51 |
| QAGNet (Deng et al., 2024) | CVPR-2024 | SOR | Swin-L | 0.771 | 0.857 | 5.78 | 0.616 | 0.818 | **6.71** |
| DSGNN (Wu et al., 2024) | CVPR-2024 | SOR | Swin-L | 0.761 | 0.856 | 5.41 | 0.602 | 0.801 | 7.01 |
| PoseSOR (Guan & Lau, 2024a) | ECCV-2024 | SOR | Swin-L | 0.667 | 0.856 | 7.87 | 0.551 | 0.812 | 7.32 |
| Ours | - | SOR | Swin-L | **0.787** | 0.869 | **5.28** | **0.624** | **0.822** | 6.89 |

images, 1,436 validation images, and 2,418 test images, with each image annotated with up to five salient instances ranked by saliency levels. The IRSR dataset includes 6,059 training images and 2,929 test images, with each image containing up to eight ranked salient instances.

**Evaluation Metrics.** We employ three widely-used evaluation metrics for the SOR task: (1) Mean Absolute Error (MAE), which measures pixel-level discrepancies between predicted saliency instance masks and ground truth annotations; (2) Salient Object Ranking (SOR) scores (Islam et al., 2018), which computes the Spearman's rank correlation coefficient to evaluate the consistency between predicted saliency rankings and ground truth rankings. This metric tends not to penalize the detection errors such as missed or false-positive instances; (3) Segmentation-Aware SOR (SA-SOR) (Liu et al., 2021a) is proposed to correct the above limitation with the SOR score by combining Pearson correlation with detection penalties. It excludes unmatched predictions (missing real objects or detecting fake ones) through instance matching and score suppression, ensuring that the ranking score reflects both detection and ordering accuracy.

## 4.1 COMPARISON TO STATE-OF-THE-ART METHODS

**Quantitative Comparison.** As shown in Table 1, we conduct a comprehensive comparison of the proposed framework with state-of-the-art methods on the standard ASSR and IRSR benchmarks. For a fair comparison, we retrain all these methods on both the ASSR and IRSR benchmarks. Our method achieves state-of-the-art SA-SOR scores while maintaining competitive advantages in both SOR and MAE. Particularly, on the ASSR dataset, when using the Swin-L backbone, our method outperforms the current best model, QAGNet (Deng et al., 2024), by 1.95% on the SA-SOR metric, while simultaneously reducing MAE by 8.65%. This validates the efficacy of our perception-viewing modeling, where text and query features are mutually influenced and updated during the cycle.

**Qualitative Comparison.** Fig. 4 demonstrates the superior segmentation and ranking performances of our model, compared to all other methods. Notably, in multiple challenging cases (e.g., $2nd$, $6th$, and $8th$ rows), our method successfully predicts the whole salient object ranks while the results of competing methods exhibit consistent error patterns. The reason is that although other methods also utilize various cues such as shape (Wu et al., 2024) and human pose (Guan & Lau, 2024a), these cues derived solely from the image itself are inherently less representative. For example, in the $4th$ row, PoseSOR incorrectly shifts the focus from the two individuals in front of the TV to the television itself based on their poses. In contrast, our method leverages the perceptual cue "video games" to first drive our model's focus to the television, and then to the participants who are playing the video games. By integrating this cognitive process, our approach becomes more broadly applicable across a variety of scenarios compared to relying solely on cues such as shapes or human pose.

Another example is shown in the $7th$ row, where the scene-perception cue "carriage" initially directs our method's focus to the leading horse, and then shifts to the people on either side, culminating in the understanding that people are riding in a carriage. These visual comparisons generally verify that

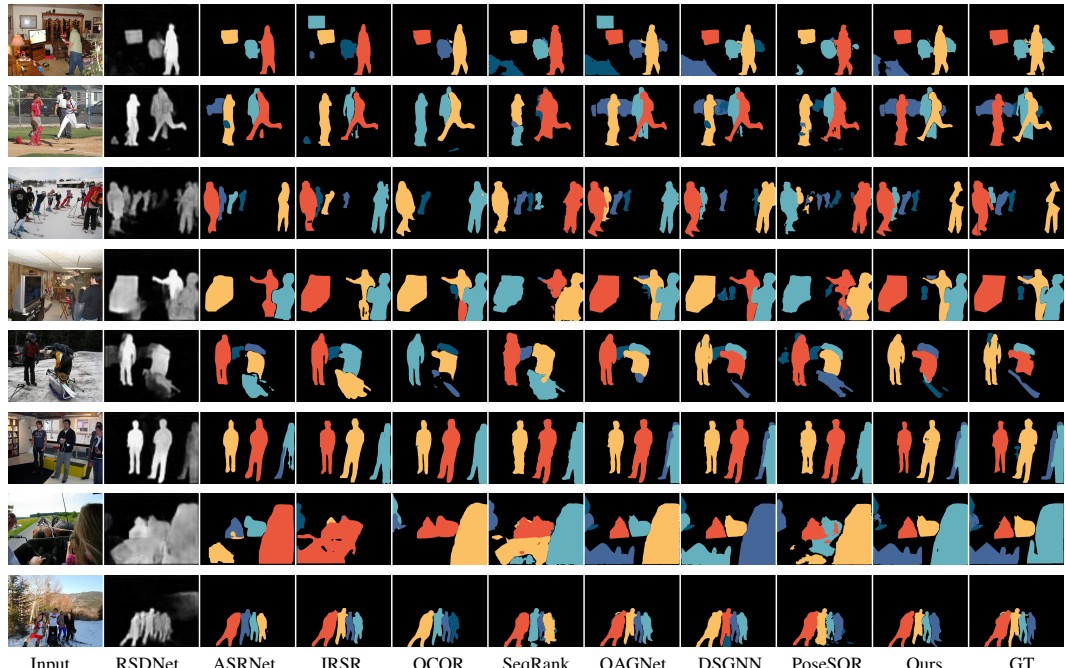

| Input | RSDNet | ASRNet | IRSR | OCOR | SeqRank | QAGNet | DSGNN | PoseSOR | Ours | GT |

Figure 4: Visual comparison between results of our method and those of eight state-of-the-art methods. Our method produces more faithful salient object ranking results.

modeling perception-viewing cycle in our approach enables reasoning the sequence of attention shifts during scene comprehension. Refer to the Appendix B for more visual comparisons.

## 4.2 INTERNAL ANALYSIS

To validate the effectiveness of each proposed module and design choice, we conduct thorough ablation studies on the ASSR benchmark.

**Analyzing the Model Components.** We first evaluate each module's efficacy on the ASSR benchmark systematically through controlled ablation studies. Table 2 shows the results. We begin with the baseline method where object queries from the query decoder are directly fed into segmentation and ranking heads (denoted as "**I**"). We then gradually introduce new components into the baseline method **I** as: adding captioning supervision (denoted as "**II**"), incorporating the CMQC module (denoted as "**III**"), exploiting the saliency reweighing and the gating

Table 2: Ablation analysis of different techniques in the proposed SP and GR modules. $\mathbf{S}^{(k)}$ denotes the saliency score of object queries in each step.

| Settings | Caption | CMQC | SITA $\mathbf{S}^{(k)}$ | SITA Gate | SA-SOR ↑ | SOR ↑ | MAE ↓ |
|---|---|---|---|---|---|---|---|
| I | - | | | | 0.697 | 0.841 | 7.71 |
| II | ✓ | | | | 0.722 | 0.847 | 6.83 |
| III | ✓ | ✓ | | | 0.729 | 0.849 | 6.62 |
| IV | ✓ | | ✓ | | 0.734 | 0.847 | 6.21 |
| V | ✓ | ✓ | | ✓ | 0.748 | 0.854 | 6.27 |
| VI | ✓ | ✓ | ✓ | ✓ | **0.752** | **0.861** | **5.99** |

mechanism separately (denoted as "**IV**" and "**V**", respectively) and jointly as the whole SITA module (denoted as "**VI**"). We can see from Table 2 that while the simple baseline method **I** may not perform well, gradually incorporating the proposed techniques (from **II** to **VI**) brings performance gains continuously under all three metrics.

**Analyzing the Numbers of Iterative Steps.** We then evaluate the impact of different numbers of iterative steps within the proposed cyclic interaction. We report the ablation results in Table 3, where "Selection of S" indicates whether saliency scores are derived from initial object queries before interaction (denoted as "First") or final refined queries after the interaction process (denoted as "Last"). By comparing settings **I** and **II**, we can see that the interaction mechanism significantly enhances the accuracy of salient object ranking, which verifies our core idea of building the cyclical perception-viewing interactions. The comparison among settings **II**, **III**, and **IV** shows that increasing

the number of interaction steps tends to produce better SOR performances and achieve the best performance when the number of steps is set to 5, while we observe the MAE tends to be saturated.

Table 3: Ablation study on the number of iterative steps. "Selection of $S$" indicates whether saliency scores are computed from object queries before ("First") or after ("Last") the interaction process.

| Settings | Steps | Selection of $S$ | | SA-SOR ↑ | SOR ↑ | MAE ↓ |
|---|---|---|---|---|---|---|
| | | First | Last | | | |
| I | 3 | ✓ | | 0.531 | 0.714 | 10.23 |
| II | 3 | | ✓ | 0.747 | 0.848 | 5.81 |
| III | 4 | | ✓ | 0.754 | 0.851 | 5.74 |
| IV | 5 | | ✓ | **0.767** | **0.856** | **5.73** |
| V | 6 | | ✓ | 0.764 | 0.854 | 5.74 |

Table 4: Ablation study on caption refinement.

| Settings | Number of Steps | Refine | | CIDEr ↑ | SPICE ↑ |
|---|---|---|---|---|---|
| | | First | Last | | |
| I | 0 | - | - | 0.362 | 0.114 |
| II | 1 | ✓ | | 0.397 | 0.125 |
| III | 1 | | ✓ | 0.416 | 0.138 |
| IV | 3 | | ✓ | 0.433 | 0.149 |
| V | 5 | | ✓ | **0.462** | **0.161** |
| VI | 6 | | ✓ | 0.457 | 0.158 |

**Analysis of Caption Refinement.** We now analyze the impact of the number of iteration steps on caption generation. The results are shown in Table 4, where "First" and "Last" represent whether the hidden states for caption generation are selected before or after the proposed interaction. We report the CIDEr and SPICE metrics for evaluating image caption quality, with CIDEr focusing on assessing content consistency and SPICE focusing on

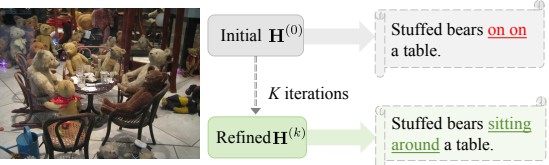

Figure 5: A comparison of a generated caption before and after refinement.

assessing the quality of semantic information. In setting **I**, we use an independent text decoder branch upon the shared backbone, which does not interact with object queries (aligned with Table 2 Setting **II**). From settings **II** to **V**, we gradually increase the number of interaction steps. The comparison between settings **I** and **II** demonstrates that the interaction can enhance the hidden state representation, while the comparison between **II** and **III** reveals the effectiveness of refining the hidden state. Settings **IV** to **VI** show that setting the number of iterations to 5 achieves the best caption generation performance, which we use in this work. We also discuss an early stopping criterion in Appendix C. Fig. 5 shows a comparison of a generated caption before and after refinement.

**Analysis of Semantic Density** In free-viewing, humans dynamically assign and adjust their attention to different objects in a way that can maximize their contextual comprehension. Our method models this cyclical interaction between scene perception and viewing for saliency ranking. Consequently, our method is expected to be more effective in semantically rich scenarios compared to simple ones. To verify this, we first define the *semantic density* $\rho$ of an image as the ratio of the number of words in the ground truth caption to the number of salient objects:

$$\rho = \text{round}(\frac{\text{Number of Words in Caption}}{\text{Number of Salient Objects}}). \tag{13}$$

Images with higher semantic density contain richer contextual information. We then randomly select 600 images from the ASSR test set and compute $\rho$ for each image. We group these images based on $\rho$ values and calculate the mean SA-SOR score for each group. The results are shown in Table 5.

Table 5: Mean SA-SOR scores grouped by semantic density $\rho$ on 600 images from the ASSR test set.

| $\rho$ | 2 | 3 | 4 | 5 | 6 | 7 | 8 | 9 | 10 | 11 | 12 | 13 | 14 | 15 |
|---|---|---|---|---|---|---|---|---|---|---|---|---|---|---|
| Mean SA-SOR | 0.796 | 0.747 | 0.774 | 0.838 | 0.819 | 0.629 | 0.725 | 0.808 | 0.90 | 0.909 | 0.880 | 0.895 | 0.944 | 1.0 |

We further compute the Pearson correlation coefficient between $\rho$ and SA-SOR across all 600 images. As shown in Table 6, $\rho$ and SA-SOR exhibit a strong positive linear relationship, indicating that our model performs better on images with higher semantic density.

**Model Efficiency Analysis** We provide a comprehensive analysis of the computational efficiency and runtime performance of our proposed method. All experiments are conducted on a single NVIDIA RTX 3090 GPU. Table 7 summarizes the inference time and frames per second (FPS) for our method

Table 6: Pearson correlation between semantic density $\rho$ and SA-SOR on the ASSR test subset.

| Dataset | Mean $\rho$ | Pearson $r$ | $p$-value |
|---|---|---|---|
| ASSR-test (600 images) | 6.07 | 0.714 | 0.00416 |

under various configurations (backbone, input resolution, and number of cyclical steps $K$), alongside comparisons with other state-of-the-art methods for reference.

Table 7: Comprehensive runtime performance (inference time / FPS) on RTX 3090.

| Method | Backbone | 512×512 | 768×768 | 1024×1024 ($K = 1$) | 1024×1024 ($K = 5$) |
|---|---|---|---|---|---|
| Ours | Swin-L | 70ms / 14.3 | 118ms / 8.5 | 191ms / 5.2 | 205ms / 4.9 |
| | Swin-B | 49.8ms / 20.1 | 80.7ms / 12.4 | 136ms / 7.3 | 148ms / 6.8 |
| PoseSOR (Guan & Lau, 2024a) | Swin-L | – | 98ms / 10.2 | 152ms / 6.6 | – |
| QAGNet (Deng et al., 2024) | Swin-L | – | 294ms / 3.4 | 384ms / 2.6 | – |

As shown in the table, the majority of the computational cost originates from the Transformer-based backbone, whose attention mechanism has a quadratic complexity with respect to spatial resolution ($\mathcal{O}((HW)^2)$). In contrast, increasing the cyclical interaction steps from $K = 1$ to $K = 5$ introduces minimal overhead, merely adding 14ms (approximately 7% at 1024×1024 for Swin-L). When configured for speed (Swin-B backbone), our method achieves interactive speeds, reaching 20.1 FPS at 512×512 and 12.4 FPS at 768×768 resolution, with a modest performance trade-off (SA-SOR decreases by 2.0% and 3.5% at 768×768 and 512×512 compared to 1024×1024, respectively). In a high-accuracy setting (Swin-L at 1024x1024), our method operates at 5.2 FPS ($K = 1$) and 4.9 FPS ($K = 5$), which is competitive among Transformer-based SOR methods.

## 5 CONCLUSION

In this paper we have proposed to model the cyclical interaction between perception and viewing for SOR. Our method introduces two key components: the Story Prediction (SP) module, which simulates human perceptual process through image caption generation, and Guided Ranking (GR) module, which predicts saliency ranking under SP's guidance. Through iterative cross-modal refinement, object queries in GR and textual features in SP interact dynamically, effectively mimicking human-like perception-viewing cycles. Extensive experiments on SOR benchmarks demonstrate the superior performance of our method. Nevertheless, our method does have limitations. When the scene semantics is weak, the guidance provided by the perceptual process of our model may be limited. As illustrated in Fig. 6, our method may fail in scenes where the saliency of objects is mainly determined by low-level features such as colors, shapes, and positions.

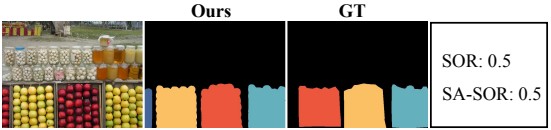

**GT caption:** A display at grocery store filled with fruits and vegetables next to jars.

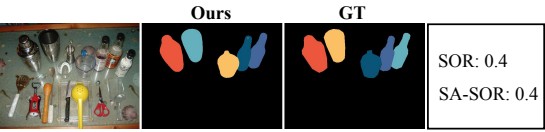

**GT caption:** An assortment of cooking utensils including a measuring cup and scissors.

Figure 6: Failure cases. Our model may fail to predict the correct saliency rank when the semantic information of the scene is relatively weak, in which case the ground truth caption (from humans) may not provide much information related to each salient object.

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

# A    THEORETICAL JUSTIFICATION OF CMQC AND SITA

## A.1    NEURAL GAIN MODULATION INTERPRETATION OF SITA

The gating mechanism in SITA (Eq. 8 and Eq. 9 of main paper) is inspired by neural gain modulation in biological vision systems (Peelen et al., 2024). The gate $\mathbf{G}$ modulates the text features $\mathbf{H}$ based on saliency-weighted visual context. This mimics how attentional gain in the brain prioritize high-confidence, salient information and suppresses irrelevant or noisy signals.

## A.2    PREDICTIVE CODING INTERPRETATION OF CMQC

Predictive coding theory posits that the brain minimizes prediction errors through iterative residual updates (Rao & Ballard, 1999). We show that the CMQC module (Eq. 10 of main paper) implements a similar mechanism for cross-modal alignment.

Let the optimal object query conditioned on textual context be denoted as $\mathbf{Q}_g^* = \mathbb{E}[\mathbf{Q}_g|H]$. The cross-attention output in CMQC approximates the prediction error:

$$\text{MultiHeadAttn}(\mathbf{Q}_g^{(k)}, H^{(k)}) \approx \alpha(\mathbf{Q}_g^* - \mathbf{Q}_g^{(k)}), \tag{14}$$

where $\alpha$ is an effective learning rate. Then the update becomes:

$$\mathbf{Q}_g^{(k+1)} = \mathbf{Q}_g^{(k)} + \alpha(\mathbf{Q}_g^* - \mathbf{Q}_g^{(k)}), \tag{15}$$

which implies:

$$\mathbf{Q}_g^{(k+1)} - \mathbf{Q}_g^* = (1 - \alpha)(\mathbf{Q}_g^{(k)} - \mathbf{Q}_g^*). \tag{16}$$

After $K$ iterations:

$$\mathbf{Q}_g^{(K)} - \mathbf{Q}_g^* \approx (1 - \alpha)^K(\mathbf{Q}_g^{(0)} - \mathbf{Q}_g^*). \tag{17}$$

As $K \to \infty$, $\mathbf{Q}_g^{(K)} \to \mathbf{Q}_g^*$, indicating that the residual update minimizes the prediction error. This process is equivalent to gradient descent on the loss:

$$\mathcal{L}_{\text{pc}} = \mathbb{E}\left[\|\mathbf{Q}_g - \mathbb{E}[\mathbf{Q}_g|H]\|^2\right], \tag{18}$$

$$\text{MultiHeadAttn}(\mathbf{Q}_g^{(k)}, H^{(k)}) \approx -\eta \nabla_{\mathbf{Q}_g^{(k)}} \mathcal{L}_{\text{PC}} = -\eta \cdot 2(\mathbf{Q}_g^{(k)} - \mathbf{Q}_g^*), \tag{19}$$

where $\eta$ is an effective learning rate and corresponds to Eq. 14. The proof above show how CMQC residual achieves predictive error minimization.

# B    MORE VISUAL RESULTS

We provide more visual results in Figure 7, where our method produces more faithful salient object ranking results compared with state-of-the-art methods.

# C    EARLY STOPPING CRITERION

Our method employs a fixed number of iterative cycles $(K = 5)$ to achieve optimal performance in saliency ranking. To accommodate latency-sensitive scenarios, we further explore an adaptive strategy that dynamically adjusts the number of iterations. Specifically, we define an adaptive stopping criterion as follows:

$$\frac{\|\mathbf{Q}_g^{(k)} - \mathbf{Q}_g^{(k-1)}\|^2}{N} < \varepsilon, \tag{20}$$

where $\varepsilon$ is a convergence threshold. When $\varepsilon = 0.05$, the average iteration count is reduced from 5 to 4.1, with approximately 86% of images converging by the fourth cycle. This results in a computational saving of approximately 4ms per frame while maintaining model accuracy, with performance degradation remaining negligible (SA-SOR drop below 0.5%).

# D    MORE ABLATION STUDIES AND EXPERIMENTS

## D.1    INFLUENCE OF THE NUMBER OF QUERIES.

We report the results of using different numbers of object queries in the Table 8. Setting the number of queries to 200 is suitable for our SOR task. Further increasing the number to 300 degrades the SOR performance as it would introduce background noise. Decreasing the number of queries to 100 also degrades the SOR performance as it impedes the model's capability to represent objects in complex scenes (for example, small or partially-occluded objects).

Table 8: Impact of query numbers on SOR performance.

| Query Numbers | SA-SOR↑ | SOR↑ |
|---|---|---|
| 100 | 0.7706 | 0.8565 |
| 200 | 0.7732 | 0.8561 |
| 300 | 0.7724 | 0.8543 |

## D.2    SCALING UP WITH MORE POWERFUL TEXT DECODER

Table 9 shows replacing the BLIP-1 text decoder (in Setting II and VI of Table 2) with a stronger text decoder (i.e., BLIP-2 OPT-2.7B) helps improve the performance.

Table 9: Comparison of different text decoders under Setting II.

| Methods | SA-SOR↑ |
|---|---|
| Setting II (Table 2 of main paper) | 0.722 |
| Setting II (BLIP-1 →BLIP-2) | 0.739 |
| Setting VI | 0.752 |
| Setting VI (BLIP-1 →BLIP-2) | 0.762 |

## D.3    SOR-GUIDED IMAGE CAPTIONING.

We further studied whether SOR could help with image captioning. We adopt a strong off-the-shelf MLLM, Qwen2.5-VL-7B, as the baseline model to generate image captions. This baseline model takes as input an image and a text prompt that asks the MLLM to generate a caption for the input image. We further construct the zero-shot SOR-guided caption generation model based on the baseline. Specifically, in addition to the input image, we also provide masks for salient objects and their saliency ranks. The prompt explains these saliency ranks to the MLLM (as a human attention-shift sequence) and asks it to generate the caption. The results (on the ASSR test set) in Table 10 show that SOR guidance brings consistent improvements across all captioning metrics, suggesting that salient object ranking can serve as a useful prior for enhancing image captioning.

Table 10: Comparison of captioning performance with and without SOR guidance.

| Method | BLEU↑ | METEOR↑ | ROUGE-L↑ | CIDEr↑ | SPICE↑ |
|---|---|---|---|---|---|
| Baseline | 0.0576 | 0.2966 | 0.2454 | 0.0883 | 0.1791 |
| SOR-guided | **0.0795** | **0.3285** | **0.2589** | **0.1395** | **0.2298** |

# E    VISUALIZATION OF INTERACTION

We visualize the cross-modal interaction and results under different values of $K$.

Figure 8a plots the evolving trajectories of the top-8 ranked queries with the increasing number of interaction steps. We can see that the beginning interactions help our model gradually identify saliency ranks (e.g., query #138 is correctly ranked at step 2; query #195 overtakes query #166). The last few interaction steps tend to refine and smooth saliency rank predictions.

In Figure 8b, we visualize the $L_2$ norm of each query ($\|Q_{g(i)}^{(t)}\|_2$), which shows that these queries are numerically stable across the interactions, with no sign of numerical explosion or collapse, indicating that the cyclic update is stable in terms of feature magnitude.

Figure 8c shows text–query attention heatmaps for the top-50 ranked queries across different interaction steps. At the beginning (Step 0), most queries only attend sparsely to tokens across the entire caption, with slightly higher responses occasionally at the beginning positions of the sentences. As the interaction proceeds (Steps 1–3), attentions are shifted gradually to focus on a small set of semantically important tokens, which tends to be stable at Steps 4–5. This indicates that the cyclic interaction effectively guides the salient instances to focus on the words that are most relevant to the scene description, rather than diffusing attention over irrelevant tokens.

Figure 9 visualizes the SOR results of an example under different numbers of interaction steps, which shows that the interactions gradually adjust the saliency ranks of the man over the TV screen.

# F    THE USE OF LLM

This research does not involve the use of Large Language Models (LLMs) in its core contributions, such as for model training or fine-tuning. LLMs were used solely for the purpose of polishing the writing of the manuscript. These uses do not affect the originality or core methodology of the research and therefore do not require detailed declaration.

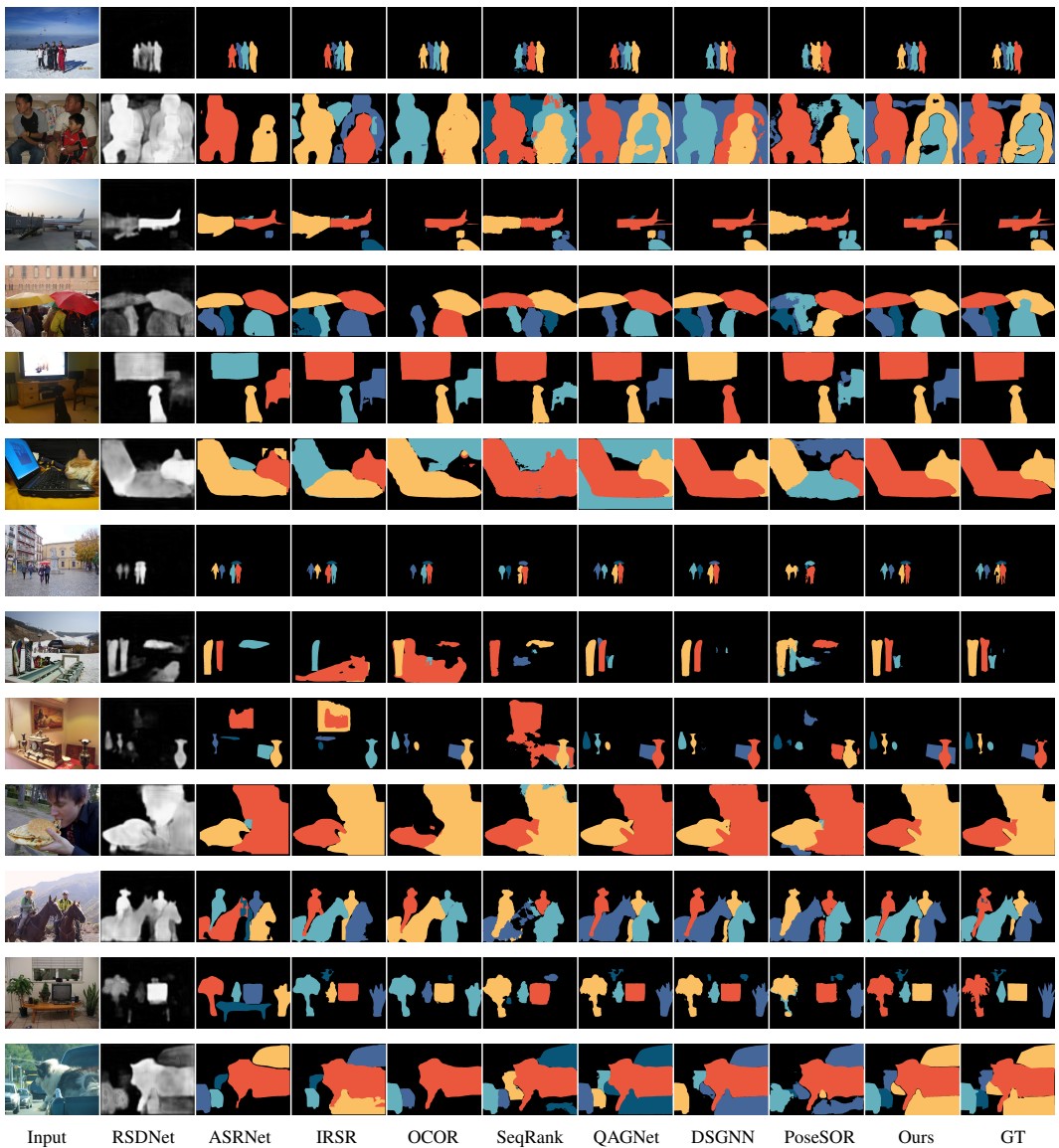

Figure 7: Visual comparisons between our method and eight state-of-the-art methods.

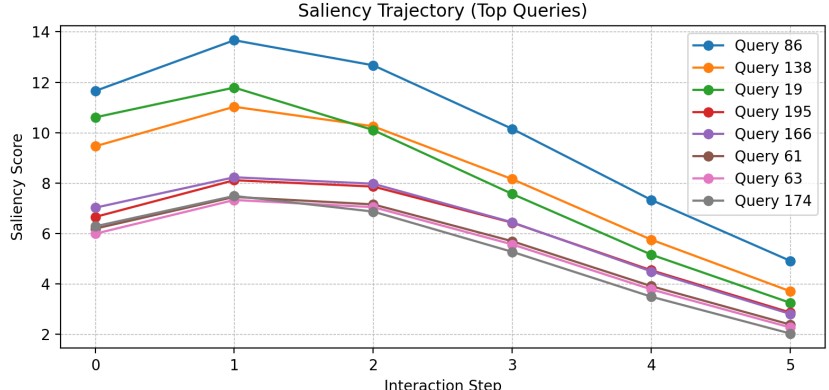

(a) Evolving trajectories of top-8 ranked queries across different interaction steps.

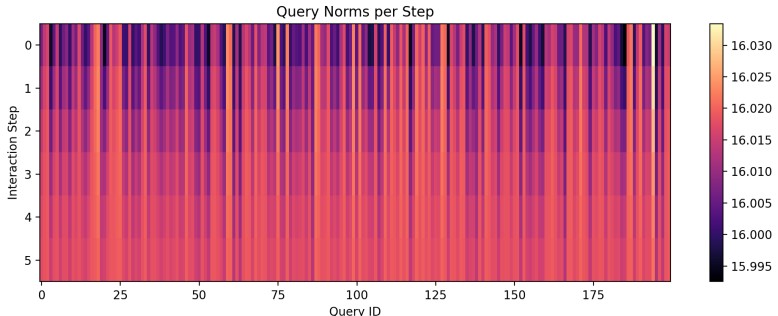

(b) $L_2$ norms of all object queries over interaction steps.

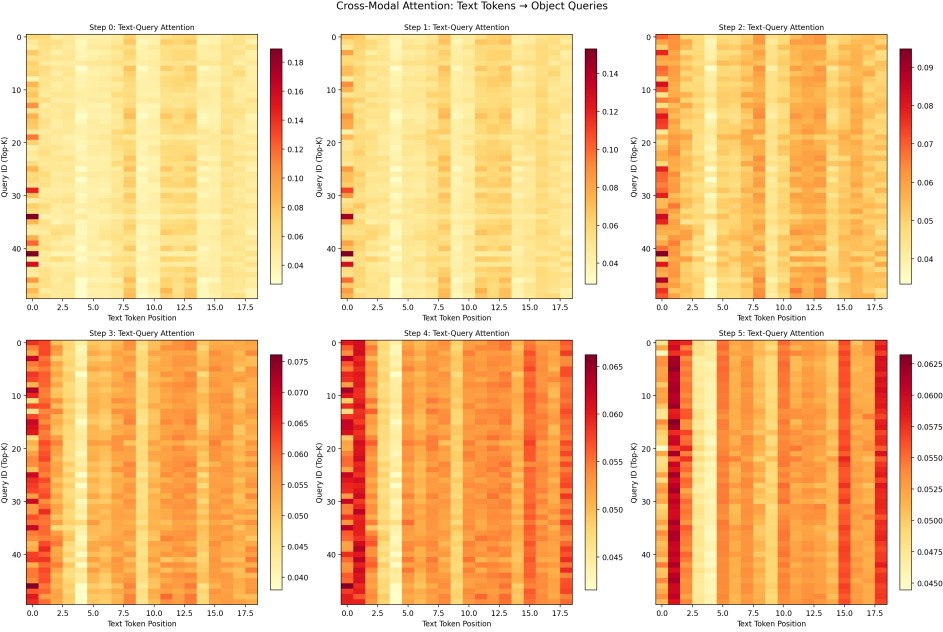

(c) Text–query attention heatmaps for the top-50 ranked queries across interaction steps.

Figure 8: Qualitative analysis of the cyclic interaction.

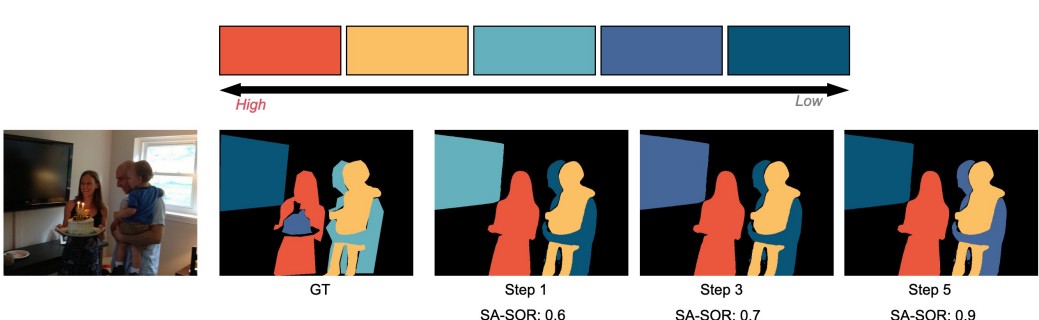

Figure 9: Visualization of results under different interaction steps.

