# OpenReview forum: "Salient Object Ranking via Cyclical Perception-Viewing Interaction Modeling"
_ICLR.cc/2026/Conference — ICLR 2026 Poster_

### Official Review · Reviewer_5n6p · 2025-10-26

**Soundness:** 2
**Presentation:** 3
**Contribution:** 2
**Rating:** 4
**Confidence:** 4

**Summary:**

This paper proposed a method for saliency object ranking. A story prediction module predicts the caption of the image and a guided ranking module predicts the saliency rankings. The cyclical interaction module aligns and refines the caption and the ranking iteratively. The experimental results seemed to show the proposed method outperformed previous SOTA.

**Strengths:**

- The cyclical interaction uses caption to guide saliency object ranking.
- Ablations shows the effectiveness of the SITA and CMQC in the proposed method.

**Weaknesses:**

- The segmentation head is unclear. The performance increase could potentially due to using a strong pretrained segmentation model.
- The retained QAGNet has lower scores across metrics compared to the ones reported in the original paper. This is critical since the results of the proposed method does not outperform the reported results of QAGNet.

**Questions:**

- Is the segmentation head a pretrained segmentation model or else? A strong segmentation model could favor the MAE.
- What is the impact of number of object queries on results? An ablation study will be beneficial to see the impact.
- Would a stronger text decoder leads to better performance?
- What is the reason of decreased performance of retrained QAGNet? Did the authors use different training details or different evaluation settings or else?
- Intuitively, the proposed method could also improve the performance on image captioning task. I am wondering if salient object ranking could help with image captioning. It will be interesting to see results compared with SOTA image captioning methods.

---

> ### Author Response · Authors · 2025-11-21
>
> Q1: The segmentation head is unclear. The performance increase could potentially due to using a strong pretrained segmentation model. Is the segmentation head a pretrained segmentation model or else? A strong segmentation model could favor the MAE.
>
>
>
> A1: The segmentation head is pretrained on the COCO segmentation dataset, following previous SOR methods, QAGNet (Deng et al., 2024) and DSGNN (Wu et al., 2024). This ensures a fair comparison with these methods including the MAE comparison.
>
>
>
> ------
>
>
>
> Q2: The retained QAGNet has lower scores across metrics compared to the ones reported in the original paper. This is critical since the results of the proposed method does not outperform the reported results of QAGNet. What is the reason of decreased performance of retrained QAGNet? Did the authors use different training details or different evaluation settings or else?
>
>
>
> A2: We follow previous methods (Guan & Lau, 2024a; Guan & Lau, 2024b) to retrain all methods for comparisons, including the QAGNet. We use the official training scripts (with default hyper-parameters setting unmodified, such as the optimizer, batch size, learning rate, and resolution) released by QAGNet for its retraining.
>
> The performance gap may be caused by two training differences: (1) we use four A100 GPUs to retrain QAGNet, while the QAGNet paper uses four A6000 GPUs; and (2) we set the random seed to 42 (for all methods) while the random seed used in the QAGNet paper is not reported.
>
>
>
> ------
>
>
>
> Q3: What is the impact of number of object queries on results? An ablation study will be beneficial to see the impact.
>
>
>
> A3: As suggested, we report the results of using different numbers of object queries in the table below.
>
> We find that setting the number of queries to $200$ is suitable for our SOR task. Further increasing the number to $300$ degrades the SOR performance as it would introduce background noise. Decreasing the number of queries to $100$ also degrades the SOR performance as it impedes the model’s capability to represent objects in complex scenes (for example, small or partially-occluded objects).
>
> | **Query Numbers** | **SA-SOR$\uparrow$** | **SOR$\uparrow$** |
> | ----------------- | -------------------- | ----------------- |
> | 100               | 0.7706               | 0.8565            |
> | 200               | 0.7732               | 0.8561            |
> | 300               | 0.7724               | 0.8543            |
>
>
>
> ------
>
>
>
> Q4: Would a stronger text decoder leads to better performance?
>
>
>
> A4: We expect that a stronger text decoder could lead to better performance. Our results (in the table below) show that replacing the BLIP-1 text decoder (in Settings II and VI of Table 2 of the main paper) with a stronger text decoder (i.e., BLIP-2 OPT-2.7B) consistently improves the performance.
>
> | **Methods**                              | **SA-SOR$\uparrow$** |
> | ---------------------------------------- | -------------------- |
> | Setting II (Table 2 of main paper)       | 0.722                |
> | Setting II (BLIP-1 $\rightarrow$ BLIP-2) | 0.739                |
> | Setting VI                                     | 0.752                |
> | Setting VI (BLIP-1 $\rightarrow$ BLIP-2)       | 0.762       |
>
>
>
> ------

---

> > ### Author Response · Authors · 2025-11-21
> >
> > Q5: Intuitively, the proposed method could also improve the performance on image captioning task. I am wondering if salient object ranking could help with image captioning. It will be interesting to see results compared with SOTA image captioning methods.
> >
> >
> >
> > A5: Thanks for the interesting question. Note that our work focuses on incorporating image captioning as a supervision for SOR. To examine whether SOR could help with image captioning, we have conducted the following experiment:
> >
> >
> >
> > We adopt a strong off-the-shelf MLLM, Qwen2.5-VL-7B, as the baseline model to generate image captions. This baseline model takes as input an image and a text prompt that asks the MLLM to generate a caption for the input image. We further construct the zero-shot SOR-guided caption generation model based on the baseline. Specifically, in addition to the input image, we also provide masks for salient objects and their saliency ranks. The prompt explains these saliency ranks to the MLLM (as a human attention-shift sequence) and asks it to generate the caption.
> >
> >
> >
> > We evaluate on the ASSR test set. The table below reports the results, which show that SOR guidance brings consistent improvements across all captioning metrics, suggesting that salient object ranking can serve as a useful prior for enhancing image captioning.
> >
> > | **Method** | **BLEU$\uparrow$** | **METEOR$\uparrow$** | **ROUGE-L$\uparrow$** | **CIDEr$\uparrow$** | **SPICE$\uparrow$** |
> > | ---------- | ------------------ | -------------------- | --------------------- | ------------------- | ------------------- |
> > | Baseline   | 0.0576             | 0.2966               | 0.2454                | 0.0883              | 0.1791              |
> > | SOR-guided | **0.0795**         | **0.3285**           | **0.2589**            | **0.1395**          | **0.2298**          |

---

> > ### Comment · Reviewer_5n6p · 2025-11-21
> >
> > Thanks for the additional experiments. They clarified most of my comments and questions. For the comparison with retrained baselines, it will be beneficial to see mean and standard deviation on all metrics with inferences with multiple random seeds.

---

> > > ### Author Response · Authors · 2025-11-29
> > >
> > > We are glad to know that our responses have addressed your previous concerns and questions. As suggested, we retrain the QAGNet (the second best performing method) and our method six times with different random seeds, and report their mean and standard deviation results on the ASSR benchmark in the table below. We can see that (1) our method consistently outperforms QAGNet on all three metrics, and (2) QAGNet tends to be sensitive to different random seeds (with a notable standard deviation of 0.22 for the MAE metric), while our method is more robust to these different random seeds. This demonstrates the superiority of the explicit establishment of cyclical interactions between scene perception and human attention shift in our method, over the reliance on bottom-up influence of image features for SOR in QAGNet.
> > >
> > > | Methods | Random Seeds | SA-SOR ↑ | SOR ↑ | MAE ↓ |
> > > |--------|--------------|----------|-------|-------|
> > > | **QAGNet** | 1 | 0.770 | 0.855 | 5.39 |
> > > |        | 2 | 0.776 | 0.855 | 5.39 |
> > > |        | 3 | 0.773 | 0.854 | 5.28 |
> > > |        | 4 | 0.767 | 0.855 | 5.89 |
> > > |        | 5 | 0.777 | 0.852 | 5.41 |
> > > |        | 6 | 0.771 | 0.859 | 5.76 |
> > > |        | **mean ± std** | **0.772 ± 0.004** | **0.855 ± 0.002** | **5.52 ± 0.22** |
> > > | **Ours** | 1 | 0.788 | 0.864 | 5.21 |
> > > |        | 2 | 0.791 | 0.857 | 5.30 |
> > > |        | 3 | 0.787 | 0.867 | 5.21 |
> > > |        | 4 | 0.785 | 0.862 | 5.30 |
> > > |        | 5 | 0.792 | 0.858 | 5.41 |
> > > |        | 6 | 0.788 | 0.866 | 5.30 |
> > > |        | **mean ± std** | **0.789 ± 0.003** | **0.862 ± 0.004** | **5.29 ± 0.07** |

---

### Official Review · Reviewer_siW4 · 2025-10-27

**Soundness:** 3
**Presentation:** 2
**Contribution:** 3
**Rating:** 6
**Confidence:** 3

**Summary:**

This paper proposes a novel framework that models the cyclical interaction between perception and viewing for the Salient Object Ranking (SOR) task. The method introduces two key components: a Story Prediction (SP) module that simulates the human perceptual process through image caption generation, and a Guided Ranking (GR) module that predicts saliency rankings under the guidance of the SP module.

**Strengths:**

（1）Novel Cognitive-Inspired Framework.
The paper introduces a cyclical perception–viewing model inspired by human visual cognition, which is strongly supported by established cognitive and psychological theories. And the introduction is easy to follow.

（2）Extensive experiments.
The paper conducts both qualitative and quantitative experiments, and also provides an analysis of inference time. Moreover, the visualized experimental results clearly and intuitively demonstrate the improvements achieved by the proposed method.

**Weaknesses:**

（1）The paper lacks a clear comparison with the recent top-down method, Language-Guided Salient Object Ranking (CVPR 2025), and its performance remains inferior to the results reported in that study.

（2）In the Method Overview section, the symbols used in the equations do not correspond to those shown in Figure 2, which makes it confusing to understand the inputs and outputs of each module.

（3）The experimental section mainly provides data and setup details but offers limited analysis or discussion to explain the observed results.

**Questions:**

(1) Explain the differences between the proposed method and Language-Guided Salient Object Ranking (CVPR 2025). Moreover, the performance of this method is still inferior to that of the existing work.

(2) In Eq.(1), when $l=1$, what dose $Q_{l-1}$ refer to?

(3) In Table 2, for Setting II (“independent caption generation”), there is a performance improvement even without interaction between caption and visual features, which is confusing. Could the authors clarify this behavior?

(4) How is the ground-truth (GT) caption obtained?

---

> ### Author Response · Authors · 2025-11-21
>
> Q1: The paper lacks a clear comparison with the recent top-down method, Language-Guided Salient Object Ranking (CVPR 2025), and its performance remains inferior to the results reported in that study. Explain the differences between the proposed method and Language-Guided Salient Object Ranking (CVPR 2025).
>
>
>
> A1: The code of LG-SOR was not released when we submitted our work. Hence, we were unable to retrain and report their results. According to the reported performance in their paper, although our method achieves relatively lower SOR and MAE results, we still obtain comparable SA-SOR results, but with a lower computational cost: LG-SOR requires an external LVLM to generate language descriptions in both training and inference stages, while our method does not. As a result, LG-SOR requires eight A100 (80 GB) GPUs for training, while our method is trained only on 4 RTX3090 GPUs.
>
>
>
> We further highlight the differences and similarity between LG-SOR and our method:
>
> - **Different motivations.**
>
>   Language-Guided SOR (LG-SOR) is motivated by the observation that LVLMs naturally produce scene descriptions where salient objects are mentioned in order, together with their semantic relations. It therefore treats the LVLM-generated description as a static high-level prior, and its text-guided modulation helps inject this external linguistic knowledge to improve SOR performance.
>
>   In contrast, our work is motivated by cognitive studies. We observe that human visual behavior and perception are tightly coupled during scene exploration. We explicitly build this cyclic interaction between viewing and perception through our newly proposed SP and GR modules.
>
> - **Different implementations.**
>
>   LG-SOR requires an external LVLM to generate a detailed scene description for each image, then injects this linguistic cue into a SOR backbone through the proposed text-guided modulation and reasoning modules.  The LVLM is not trained jointly with the SOR network, and the language description is a pre-computed, static signal that is fused with visual features only once before predicting the final ranks.
>
>   In contrast, our method does not rely on any pre-generated textual cues. Instead, the Story-Prediction (SP) and Guided-Ranking (GR) modules interact iteratively and jointly to produce both the caption and the final ranking. In this way, the ranking head learns the ordering of objects as an integral part of constructing scene understanding, rather than merely consuming a static caption produced by an external LVLM.
>
> - **Similarity.**
>
>   The LG-SOR can be viewed as a special case of our formulation, where a fixed description generated by a powerful LVLM guides the SOR without further interactions.  We believe that our text decoder could be replaced by a more powerful LVLM to achieve stronger results.  We have conducted preliminary experiments based on our baseline models (Setting II and Setting VI in Table 2 main paper), in which we replace our current BLIP-1 text decoder with the more powerful BLIP-2 text decoder. The table below shows a clear and consistent performance gain.
>
>
> | **Methods**                             | **SA-SOR$\uparrow$** |
> | --------------------------------------- | -------------------- |
> | Setting II (Table 2 of main paper)      | 0.722                |
> | Setting II (BLIP-1 $\rightarrow$BLIP-2) | 0.739                |
> | Setting VI                                    | 0.752                |
> | Setting VI (BLIP-1 $\rightarrow$BLIP-2)       | 0.762        |
>
> ---
>
> Q2: In the Method Overview section, the symbols used in the equations do not correspond to those shown in Figure 2, which makes it confusing to understand the inputs and outputs of each module.
>
> A2: Thanks for the comment. As suggested, we have revised Figure 2 to improve its clarity.

---

> ### Author Response · Authors · 2025-11-21
>
> Q3: The experimental section mainly provides data and setup details but offers limited analysis or discussion to explain the observed results.
>
>
> A3: Thanks for the comment. As suggested, we provide a more detailed discussion as follows.
>
> In the experimental section, we have demonstrated better results of our method over previous methods both visually and quantitatively. A key observation from the visual comparisons (Fig. 4, main paper) is that previous methods tend to over-prioritize humans against stuff, while our method can overcome this limitation, correctly ranking stuff with a significant contribution to the caption perception by learning to summarize the scene. A main observation from the quantitative comparisons (Table 1, main paper) is that many existing methods (e.g., OCOR (Tian et al., 2022a) and PoseSOR (Guan \& Lau, 2024a)) tend to mis-detect salient instances (as revealed by their high SOR but low SA-SOR results), while our method can correctly detect and rank salient objects to achieve both high SA-SOR and SOR scores, through the global guidance of scene perception.
>
> In our internal analysis, while we demonstrate that a fixed number of iterative cycles (K = 5) achieves optimal SOR performance, we further observe that our method may converge in earlier iteration steps, which inspires us to design an early stopping criterion to maintain model accuracy and reduce the computational cost (Appendix D). Moreover, we observe that our method tends to fail in weak-semantic scenes. To quantitatively study this limitation, we additionally define ``semantic density'' of scenes and show that our method performs proportionally to the density of scene semantics (Appendix E).
>
> We will include these discussions in our final revision.
>
>
>
> ------
>
>
>
> Q4: In Eq.(1), when $l=1$, what dose $Q_{l-1}$ refer to?
>
>
>
> A4: When $l = 1$, $Q_{l-1}$ refers to $Q_0$, the initial set of learnable object queries (L195, page 4).
>
>
>
> ------
>
>
>
> Q5: In Table 2, for Setting II (“independent caption generation”), there is a performance improvement even without interaction between caption and visual features, which is confusing. Could the authors clarify this behavior?
>
>
>
> A5: In Setting II, although there are no explicit interactions between the two branches, they share the same visual backbone and are optimized jointly (an additional captioning loss is imposed on the SP branch). The performance gain demonstrates that additional text modality supervision is beneficial to the saliency ranking task. We understand that “independent caption generation” can be confusing, and we have corrected it to “captioning supervision”.
>
>
>
> ------
>
>
>
> Q6: How is the ground-truth (GT) caption obtained?
>
>
>
> A6: Both ASSR and IRSR datasets are constructed based on the MS-COCO dataset, where each image is paired with 5 human-annotated captions. For each image, we randomly sample one out of five captions as the GT caption.
>
>
>
> ------

---

> ### Comment · Reviewer_siW4 · 2025-11-26
> **Offcial Comment by Reviewer siW4**
>
> Thank you for your answer to Q1, which clarified my confusion. I believe computational cost is an important comparison metric. Thank you also for answering the other questions. I will keep my score, but I will increase my confidence in the score.

---

> > ### Author Response · Authors · 2025-11-29
> >
> > We are glad to know that our responses have clarified your confusion. We will incorporate our responses and the suggested computational cost comparison in our revision. We thank this reviewer for the suggested improvements.

---

### Official Review · Reviewer_oTJC · 2025-11-02

**Soundness:** 3
**Presentation:** 3
**Contribution:** 3
**Rating:** 6
**Confidence:** 4

**Summary:**

The authors propose a Salient Object Ranking (SOR) approach that consists of two modules: the Guided Ranking (GR) module and the Story Prediction (SP) module, whose interaction enhances the overall performance of SOR.

**Strengths:**

The design of this model aligns well with the human cognitive system, such as predictive coding, and the English writing is good and clear.

**Weaknesses:**

Some experiments and details are not clearly explained. For example, in the experimental section, how was the choice of 24,000 epochs determined, and why such a large number? Could this lead to overfitting?

In addition, it would be helpful to qualitatively present the interaction between object queries and text features, as well as the results under different values of K.

**Questions:**

What training data are used for the segmentation head? Was it pre-trained on the COCO segmentation dataset? If it was trained only on the SOR dataset, would its segmentation generalization ability be affected?

Does the random selection of captions influence the results? It is recommended to include a discussion—for example, are the salient objects in the image always located in the main subject position described in the caption?

---

> ### Author Response · Authors · 2025-11-21
>
> Q1: Some experiments and details are not clearly explained. For example, in the experimental section, how was the choice of 24,000 epochs determined, and why such a large number? Could this lead to overfitting?
>
> A1: Thank you for the careful reading. We train our model for 24,000 iterations
> (with a batch size of 4), not epochs.
> This setting follows the common practice in prior SOR works (Deng et al., 2024; Wu et al., 2024; Guan \& Lau, 2024a; Guan \& Lau, 2024b), which typically set it to around 30,000 iterations. We apologize for the typo and have corrected it in the revision. We have gone through our paper and revised other typos that we can find.
>
> ---
>
>
>
> Q2: In addition, it would be helpful to qualitatively present the interaction between object queries and text features, as well as the results under different values of K.
>
> A2: As suggested, we visualize the cross-modal interaction and the results under different values of K in Figure 8 and Figure 9 (Appendix H), respectively.
>
> Figure 8(a) plots the evolving trajectories of the top-8 ranked queries with an increasing number of interaction steps.
> We can see that the beginning interactions help our model gradually identify saliency ranks (e.g., query \#138 is correctly ranked at step 2; query \#195 overtakes query \#166). The last few interaction steps tend to refine and smooth saliency rank predictions.
>
> In Figure 8(b), we visualize the $L_2$ norm of each query
> ($\||{Q_g}_{(i)}^{(t)}\||_2$), which shows that these queries are numerically stable across the interactions, with no sign of numerical explosion or collapse, indicating that the cyclic update is stable in terms of feature magnitude.
>
> Figure 8(c) shows text–query attention heatmaps for the top-50 ranked queries across different interaction steps. At the beginning (Step 0), most queries only attend sparsely to tokens across the entire caption, with slightly higher responses occasionally at the beginning positions of the sentences. As the interaction proceeds (Steps 1–3), attentions are shifted gradually to focus on a small set of semantically important tokens, which tends to be stable at Steps 4–5. This indicates that the cyclic interaction effectively guides the salient instances to focus on the words that are most relevant to the scene description, rather than diffusing attention over irrelevant tokens.
>
> Figure 9 visualizes the SOR results of an example under different numbers of interaction steps, which shows that the interactions gradually adjust the saliency ranks of the man over the TV screen.
>
> ---
>
> Q3: What training data are used for the segmentation head? Was it pre-trained on the COCO segmentation dataset? If it was trained only on the SOR dataset, would its segmentation generalization ability be affected?
>
> A3: Yes, our segmentation head is pretrained on the COCO segmentation dataset following previsous SOR methods, e.g., QAGNet (Deng et al., 2024) and DSGNN (Wu et al., 2024),
> before it is fine-tuned jointly with the entire model on the SOR datasets.
>
> ---
>
> Q4: Does the random selection of captions influence the results? It is recommended to include a discussion—for example, are the salient objects in the image always located in the main subject position described in the caption?
>
> A4: We clarify that each training image of the COCO dataset contains five corresponding captions, and we randomly select one for the caption supervision. From our experiments, this strategy does not affect the results, as all these captions were not annotated to describe all the salient objects in detail.
> Despite their differences, all captions (for a particular image) focus on summarizing the main scene context, covering salient objects necessary to describe the scene. For example, when describing a classroom image, one caption says  "*A group of people are shown standing in a classroom*", while another one says "*Two guys are standing at the front of a classroom*".
> This shows that these captions can generally provide global and consistent semantic guidance for our model.

---

### Author Response · Authors · 2025-11-21

We thank the reviewers for their constructive comments. We are glad to see reviewers' positive comments: designs aligned well with the human cognitive system (Reviewer **oTJC**), novel Cognitive-Inspired Framework and extensive experiments (Reviewer **siW4**), and the effectiveness of the proposed module (Reviewer **5n6p**).
We address the raised concerns and questions below. We will revise our paper according to all comments.

---

### Author Response · Authors · 2025-12-04
**Author Final Remarks**

Dear AC and the Reviewers,



We thank the AC for handling our submission, and the reviewers for their valuable comments and constructive suggestions. We are glad to see that, from the initial reviews, reviewers commented: our designs aligned well with the human cognitive system (Reviewer **oTJC**), novel cognitive-inspired framework and extensive experiments (Reviewer **siW4**), and effectiveness of the proposed module (Reviewer **5n6p**). We are happy to know that our previous rebuttal responses have clarified Reviewer **siW4**’s confusion and Reviewer **5n6p**’s comments/questions.

We summarize how we have addressed the raised concerns below, for AC’s reference:

1. **Implementation Details Clarification.**

   As suggested by Reviewers **oTJC** and **siW4**, we have clarified the training protocol of our model, specified the pretraining setup, and detailed our caption sampling strategy.

2. **More Ablations.**

   As suggested by Reviewers **siW4** and **5n6p**, we have additionally studied the impact of different numbers of object queries (Table 9), which confirms that our choice achieves a good balance. We have also explored using stronger text decoders with promising results (Table 10), which demonstrates the scaling-up capability of our method.

3. **Visualization of Cyclical Interaction.**

   As suggested by Reviewer **oTJC**, we have added visualizations of SP–GR interactions and different interaction steps (Fig. 8, 9), which help enhance the transparency and interpretability of our method.

4. **More Analysis.**

   As suggested by Reviewer **siW4**, we have added more discussion on the results and discussed the relations with the LG-SOR method (while LG-SOR differs from ours in terms of motivation and implementation, it can be considered as a special case of our formulation). As suggested by Reviewer **5n6p**, we have reported the mean and deviation results of QAGNet and ours, which show that our method is more robust to the random seeds and performs consistently better than QAGNet.

5. **Broader Impact.**

   As suggested by Reviewer **5n6p**, we have demonstrated that SOR can enhance the captioning ability of strong MLLMs (i.e., Qwen2.5-VL-7B in this case) in a zero-shot manner (Table 11).

Thank you for your consideration.

Regards,

Authors

---

### Meta-Review · Area_Chair_Hoa7 · 2026-01-06

**Summary:**

The initial reviews contain two positive review scores (6, 6) and one negative score (4). Particularly, the major concerns of reviewer 5n6p who gave score 4 are:
1) The segmentation head is unclear.
2) The retained QAGNet has lower scores across metrics compared to the ones reported in the original paper.
3) Some ablation studies

After reviewing the whole rebuttal, I believe all the concerns from reviewer 5n6p are resolved, and the concerns from other reviewers are mostly resolved too. So I will suggest accepting the paper.

**Reviewer Concerns:**

The most concerns are the segmentation head and the the retained QAGNet has lower accuracy compared to the original paper, both of these concerns are well resolved by the authors.

**Reviewer Scores:**

I believe reviewer 5n6p should raise their scores.

---

### Decision · Program_Chairs · 2026-01-26

Accept (Poster)